# Learning Interpretable Style Embeddings via Prompting LLMs

**Ajay Patel**
University of Pennsylvania
ajayp@upenn.edu

**Delip Rao**
University of Pennsylvania
deliprao@gmail.com

**Ansh Kothary**
Columbia University
ank2145@columbia.edu

**Kathleen McKeown**
Columbia University
kathy@cs.columbia.edu

**Chris Callison-Burch**
University of Pennsylvania
ccb@upenn.edu

## Abstract

Style representation learning builds content-independent representations of author style in text. To date, no large dataset of texts with stylometric annotations on a wide range of style dimensions has been compiled, perhaps because the linguistic expertise to perform such annotation would be prohibitively expensive. Therefore, current style representation approaches make use of unsupervised neural methods to disentangle style from content to create style vectors. These approaches, however, result in uninterpretable representations, complicating their usage in downstream applications like authorship attribution where auditing and explainability is critical. In this work, we use prompting to perform stylometry on a large number of texts to generate a synthetic stylometry dataset. We use this synthetic data to then train human-interpretable style representations we call LISA embeddings. We release our synthetic dataset (STYLEGENOME) and our interpretable style embedding model (LISA) as resources.

## 1 Introduction

Style representation learning aims to represent the stylistic attributes of an authored text. Prior work has treated the style of a text as separable from the content. Stylistic attributes have included, but are not limited to, linguistic choices in syntax, grammar, spelling, vocabulary, and punctuation (Jafari-tazehjani et al., 2020). Style representations should represent two texts with similar stylistic attributes more closely than texts with different attributes independent of what content is present in the texts.

Stylometry, the analysis of style, applies forensic linguistics to tasks like authorship attribution. Stylometry often relies on semi-manual analysis by forensic linguistic experts (Mosteller and Wallace, 1963; Holmes, 1994; Rosso et al., 2016). Computational stylometry often uses rule-based approaches utilizing count-based features like the frequencies of function words (Stamatatos, 2009; Koppel et al.,

*"Really really cool action movie imo. Casting is great too. It's like they were all born for their role in that movie."*

↓

**LISA**

↓

768-dimensional interpretable style vector

$$[\ 0.889\ ...,\ 0.613,\ ...,\ 1.000,\ ...,\ 0.949\ ]$$

The author is repeating words.

The author is complimentary.

Figure 1: An example of a 768-dimensional interpretable style vector produced by LISA, trained using a GPT-3 annotated synthetic stylometery dataset.

2009; Tausczik and Pennebaker, 2010). More modern, neural approaches attempt to learn style representations in an unsupervised fashion through a proxy task like style transfer (Shen et al., 2017; Fu et al., 2018; John et al., 2019; Dai et al., 2019; Li et al., 2019; Yi et al., 2021; Zhu et al., 2022) or authorship verification (Boenninghoff et al., 2019; Hay et al., 2020; Zhu and Jurgens, 2021; Wegmann et al., 2022). These stronger neural approaches, unlike simpler frequency-based techniques, are uninterpretable. This makes it difficult to effectively analyze their representations and their failure modes, and precludes their usage in real-world authorship attribution scenarios because interpretability and verification is critical for legal admissibility (Tiersma and Solan, 2002).

With this motivation, we propose a human-interpretable style representation model $\mathcal{M}$ which, for a given text $t$, produces a $D$-dimensional vector $\mathcal{M}(t) \in [0.0, 1.0]^D$. Each dimension corresponds to one of $D$ style attributes $\{a_0, a_1, \ldots, a_D\}$. Each element at dimension $d$ of this vector is constrained in the range $[0.0, 1.0]$ to represent the probability of the corresponding style attribute $a_d$ being present

in the text $t$. See Figure 1 for a visualization of a result from our final trained model with $D = 768$ dimensions. An immediate obstacle to train such a model is that no large dataset of texts with stylometric annotations currently exists; annotating a large number of texts on a wide variety ($D = 768$) of stylistic attributes would likely require annotators with linguistic expertise and be prohibitively expensive. Given this, we use GPT-3 (Brown et al., 2020), a large language model (LLM), and zero-shot prompts to generate a synthetic dataset we call STYLEGENOME of human-interpretable stylometric annotations for various texts. Our approach is motivated by recent works showing models trained on synthetic datasets annotated by prompting LLMs can match and sometimes even outperform models trained on human-labeled datasets (Wang et al., 2022; Gilardi et al., 2023; Huang et al., 2022; Honovich et al., 2022). Training on STYLEGENOME, we develop the **L**inguistically-**I**nterpretable **S**tyle **A**ttribute (LISA) embedding model. We summarize our primary contributions:

1. We outline an unsupervised method for producing interpretable style embeddings using zero-shot prompting and distillation.

2. We generate and release STYLEGENOME, a synthetic stylometry dataset with ~5.5M examples, the first large-scale dataset with texts paired with wide range of stylometric annotations.

3. We train, evaluate, and release LISA, the first-ever interpretable style embedding model with a wide variety of linguistic dimensions ($D = 768$). We find LISA matches the performance of existing style representations, while allowing for explainability and transparency.

## 2   Generating STYLEGENOME

To create STYLEGENOME for training LISA, we select Reddit data from the Million User Dataset (MUD) (Khan et al., 2021; Andrews and Bishop, 2019) to stylometrically annotate following prior work that uses Reddit to source a diversity of styles from different authors (Wegmann et al., 2022). We sample 10 random posts per author for 1,000 random authors, resulting in 10,000 total posts selected for annotation. We display some of the diversity of styles captured in the following examples from our Reddit authors. They vary in formality, punctuation, emoji usage, etc.:

> **Reddit User #47:** forreal though sell that **** its worth like 650$
> **Reddit User #205:** This was amazing :D Please, make more!
> **Reddit User #216:** I thought that was Israel...or Saudi Arabia...or Egypt...Venezuela?
> **Reddit User #356:** I wouldn't watch a football game like that either. It's novel, but I think its appeal is limited and again, not the same thing as actual VR where things feel tangible and to scale.

We use GPT-3[1] (Brown et al., 2020; Ouyang et al., 2022) to annotate each post in two stages.

**Stage 1**   We elicit the model with a zero-shot prompt to generate a description of the style of a given Reddit post:

> **Prompt:** Write a long paragraph describing the unique grammar style of the following passage without referring to specifics about the topic.
>
> Passage: *Wow that's really cool. I'll have to watch for that next time I watch it.*
>
> Description:

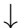

> **Generation:** The passage reflects a conversational style in its grammar, with short, simple sentences that are easy to understand. The language is informal and direct, with the speaker expressing their enthusiasm for the topic in a straightforward manner. The speaker also uses contractions, such as "I'll," to emphasize their excitement. The speaker also uses a casual tone, with the phrase "really cool" to emphasize their interest in the topic. The speaker also uses the present tense to express their anticipation for the future, with the phrase "next time I watch it." This style of grammar is effective in conveying the speaker's enthusiasm and interest in the topic.

**Stage 2**   We use another zero-shot prompt to standardize the generations from Stage 1 into short, declarative sentences with a uniform structure:

> **Prompt:** Here's a description of an author's writing style for a passage: *The passage utilizes a conversational style of grammar, with short...*
>
> Rewrite this description as a long list of short sentences describing the author's writing style where each sentence is in the format of "The author is X." or "The author uses X.".
>
> Output:

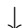

---

[1]We use `text-davinci-003` with a `temperature` setting of 0.0 and a `top_p` setting of 1.0.

> **Generation:** The author is using a conversational style of grammar.
> The author is using short, simple sentences.
> The author is using language that is informal and direct.
> The author is expressing enthusiasm for the topic in a straightforward manner.
> The author is using contractions, such as "I'll".
> The author is using a casual tone.
> The author is emphasizing their interest in the topic with the phrase "really cool".
> The author is using the present tense to express anticipation for the future.

We run both stages with 93 different Stage 1 prompts for all 10,000 posts.[2] 6 of the 93 total prompts are open-ended prompts that elicit descriptions of a passage on a broad dimension of style:

1. *Write a long paragraph describing the unique **grammar style** of the following passage...*

2. *Write a long paragraph describing the unique **vocabulary style** of the following passage...*

3. *Write a long paragraph describing the unique **punctuation style** of the following passage...*

4. *... see all 6 open-ended prompts in Appendix A.1*

The remaining 87 prompts target narrow and specific dimensions of style:

1. *Write a description of whether the author of the following passage has any **figurative language** ...*

2. *Write a description of whether the author of the following passage has any **swear words** ...*

3. *Write a description of whether the author of the following passage has any **repeated words** ...*

4. *... see all 87 targeted prompts in Appendix A.2*

The 87 targeted prompts are derived from surveys of stylometry literature, and they cover all of (Tausczik and Pennebaker, 2010)'s linguistic and psychological categories. See Appendix A.2 for more details. We report the results of an ablation experiment between the two Stage 1 prompt categories in Appendix C. Appendix D details dataset annotation costs.

**STYLEGENOME** The output of Stage 2 is sentence tokenized[3] and filtered to keep only sentences beginning with "The author". We refer to these sentences as human-interpretable *style attributes*. Our method annotates the texts with nearly 1.3M style attributes. These style attributes are represented in natural language so "The author creates a conversational tone" and "The author has a conversational tone" are counted separately in the raw dataset. Our training procedure in Section 3.1 is able to train directly on these natural language style attributes, obviating a normalization step. Some annotations may be hallucinated resulting in a noisy dataset, but we choose to train on the full synthetic dataset, without manual intervention, to maintain an unsupervised procedure following prior work (Wang et al., 2022). We hypothesize our model will find signal in the noise, which we evaluate in Section 4. The final dataset statistics can be found in Table 1.

| | |
|---|---:|
| **# of Reddit Authors** | 1,000 |
| **# of Reddit Posts** | 10,000 |
| **# of Interpretable Style Attributes** | 1,255,874 |
| **# of (Text, Style Attribute) labeled pairs** | 5,490,847 |

Table 1: Statistics for the STYLEGENOME dataset.

## 3 Method

We first distill stylometric annotation knowledge from GPT-3 into a **S**tyle **F**eature **A**greement **M**odel (SFAM). Given a text $t$ and a style attribute $a$ as input, $\text{SFAM}(t, a)$ produces an *agreement score* between 0.0 and 1.0 representing the probability of the style attribute being present in the text. By selecting a set of $D$ style attributes $\{a_0, a_1, \ldots, a_D\}$, we can use SFAM to construct LISA, our interpretable style representation model that produces $D$-dimensional vectors:

$$\mathcal{M}_{\text{LISA}}(t) = \big(\text{SFAM}(t, a_0), \text{SFAM}(t, a_1), \ldots, \text{SFAM}(t, a_D)\big)$$

The Euclidean distance between style vectors for two texts $\|\mathcal{M}_{\text{LISA}}(t_2) - \mathcal{M}_{\text{LISA}}(t_1)\|_2$ would not be particularly meaningful. We can multiply a trained weight vector $w$ or a weight matrix $W$ to the style vectors, that act as simple interpretable embedding layers. This operation would make the Euclidean distance more meaningful, for example $\|\mathcal{M}_{\text{LISA}}(t_2) * w - \mathcal{M}_{\text{LISA}}(t_1) * w\|_2$. We call the result of a LISA style vector multiplied by $w$ or $W$ a LISA style *embedding*. We discuss training in detail next, leaving hyperparameter and implementation specifics in Appendix E.

### 3.1 SFAM

We use distillation (Ba and Caruana, 2014) to teach the stylometric annotation capabilities of GPT-3 to EncT5[4] (Liu et al., 2021; Raffel et al., 2020), a smaller, more efficient student model.

---

[2] We preprocess posts to the first 25 sentences and use the `emoji` Python package to convert emojis to textual representations for better tokenization.

[3] We use the `sentence-splitter` Python package.

[4] We use the `t5-base` model and truncate at 512 tokens.

| Type | Dataset | Style Attribute | Spearman Correlation ($\rho$) |
|---|---|---|---|
| Formality | Formality in Online Communication
Grammarly's Yahoo Answers Formality Corpus | The author uses informal language. | 0.599
0.200 |
| Sentiment | Yelp Reviews
IMDB Large Movie Review Dataset
...abbreviated for space, see Appendix F for full results | The author uses a negative tone. | 0.788
0.665 |
| Emotion | DAIR.AI Emotion (Love vs. Anger)
DAIR.AI Emotion (Joy vs. Sad)
GoEmotions (Love vs. Anger)
GoEmotions (Joy vs. Sadness)
GoEmotions (Disgust vs. Desire)
...abbreviated for space, see Appendix F for full results | The author is expressing {{emotion}}. | 0.542
0.531
0.769
0.639
0.630 |
| Author Profiling | Political Slant | The author is a Democrat. | 0.005 |
| | Twitter User Gender Classification | The author is female. | 0.166 |
| | African-American Vernacular English | The author uses African-American Vernacular English. | 0.238 |
| | Shakespeare (Early Modern English) | The author uses Early Modern English. | 0.108 |
| | Wikipedia Bias | The author has a biased point of view. | 0.014 |
| Harmful Speech | HateSpeech18
Offensive Social Media | The author's writing contains hate speech.
The author uses offensive language. | 0.229
0.401 |
| Text Simplification | Simple Wikipedia
ASSET | The author uses simple language. | 0.043
0.050 |
| Linguistic Acceptability | CoLA
BLiMP | The author uses incorrect grammar. | 0.078
0.020 |
| **Average** | | | **0.342** |

Table 2: Correlation of agreement scores produced by SFAM against human judgments on texts over a wide variety linguistic and authorship dimensions. The natural language style attributes used as input to SFAM when producing the agreement scores for each dataset are also provided.

**Sampling Batches** We train EncT5 with a binary classifier head on randomly sampled batches of examples $(x_i, y_i)$ where each batch contains an equal number of positive ($y_i = 1$) and negative ($y_i = 0$) examples. The input $x_i$ consists of a style attribute $a$ and an author's text $t$ concatenated in a string "{{a}}‖{{t}}", for example $x_i =$ "The author is using a positive tone.‖You got this ;)". Labeled pairs from STYLEGENOME are sampled as positive examples such that each style attribute is sampled with equal probability. For each positive example, we perform negative sampling and retrieve a negative example text where the positive example's style attribute is likely not present. To do this, we find the 10,000 most dissimilar style attributes to the positive example's style attribute with SBERT[5] similarity. We select a text that is positively labeled with a randomly selected dissimilar style attribute as the negative example text.

**Training and Inference** Training over the ~1.3M unique style attributes in STYLEGENOME, our training dataset for SFAM is effectively a multitask mixture. Style attributes presented in natural language to the model allows the pre-trained T5 encoder to jointly learn between style attributes

and generalize to unseen style attributes using the semantic information in those natural language descriptions. This is especially desirable since some style attributes only have a handful of text examples, while others may have thousands. This setup resembles the multitask mixture trained on in Raffel et al. (2020). To validate training, we hold-out 50 random style attributes that have between 30-50 examples each as a validation set. We validate learning during training by measuring the ability of SFAM to generalize and produce accurate agreement scores for the unseen style attributes. At inference, we softmax the binary class logits to interpret them as probabilities and we take the probability of $y_i = 1$ as the agreement score. We also study the effect of the size of STYLEGENOME on performance and find that as the synthetic dataset grows, validation performance improves and SFAM generalizes to better predict agreement scores for unseen style attribute and text pairs (see Appendix B).

### 3.2 LISA Style Vectors

As discussed earlier, SFAM is directly used to produce the LISA interpretable style vectors. We arbitrarily choose $D = 768$ in this work, following the dimensionality of prior style vectors and BERT (Devlin et al., 2019). We now detail how we select the style attributes associated with each dimension $\{a_0, a_1, \ldots, a_{768}\}$ with little manual in-

---
[5]We use the `nli-distilroberta-base-v2` model (Sanh et al., 2019; Liu et al., 2019; Devlin et al., 2019; Reimers and Gurevych, 2019) for all SBERT usage in this paper.

| Text | Top 5 LISA Vector Dimensions |
|------|------------------------------|
| Lol right on | 1. 1.00 – The author is being polite.
2. 1.00 – The author is writing in a cheerful manner.
3. 1.00 – The author is using a lighthearted tone.
4. 1.00 – The author is laughing.
5. 1.00 – The author is complimentary. |
| There is the Toyota GT86 R3 ;) http://www.toyota-motorsport.com/motorsport/downloads/com_droppics/59/DSF4311.jpg | 1. 0.99 – The author is simply describing a product. *
2. 0.99 – The author is providing a visual cue to the reader.
3. 0.98 – The author simply provides information.
4. 0.97 – The author is using an emoji.
5. 0.97 – The author provides information. |
| Every time i watched this episode as a kid i was always like "WTF, JAMES, A POKEBALL ISNT EVEN A POKE-MON?! GET YOUR ACT TOGETHER, SON!!" | 1. 1.00 – The author is discussing a television show. *
2. 0.99 – The author has a hint of nostalgia.
3. 0.99 – The author is making a humorous comment.
4. 0.99 – The author is animated in their writing.
5. 0.99 – The author is laughing. |
| 13 POT MORDE BABY WOOOOOOOOOOOOOOOOO | 1. 1.00 – The author is using an elongated word.
2. 1.00 – The author is using only English words.
3. 1.00 – The author is using a single word.
4. 1.00 – The author is using swear words.
5. 1.00 – The author uses two exclamation marks. |
| No wonder everyone resorts to performing a murder spree eventually. | 1. 1.00 – The author is scornful.
2. 1.00 – The author is ungenerous.
3. 0.99 – The author is expressing antisocial behaviors.
4. 0.99 – The author is uncaring.
5. 0.99 – The author is dramatic. |
| Podcast originally refers to an iPod, and before that there was definitely TWiT, which still calls itself a Netcast | 1. 0.97 – The author uses a variety of words to describe the same concept.
2. 0.97 – The author is simply describing a product. *
3. 0.95 – The author uses specific terms related to the topic.
4. 0.94 – The author has a deep understanding of the topic.
5. 0.94 – The author is using words focusing on the past. |

Table 3: The five highest scoring dimensions from the 768-dimensional LISA vector produced on various Reddit texts. The interpretable style attribute corresponding to each dimension is displayed along with the score. We manually inspect the top style attributes and annotate them as reasonable, plausible, or incorrect. Attributes annotated with * blur the line between style and content. Error analysis can be found in Section 4.1.

tervention. The first 87 attributes $\{a_0, a_1, \ldots, a_{86}\}$ directly correspond to the features of our 87 targeted prompts in the form of "The author is using {{targeted_feature}}". The remaining 681 are downselected from the ~1.3M style attributes with filtering heuristics, choosing those that appear for at least 10 authors, but no more than 600 authors (attempting to select for frequent, yet discriminative attributes). Once a style attribute is selected to be part of the 768, we do not select another style attribute with SBERT cosine similarity $> 0.8$ to largely avoid near-duplicates. We also reject style attributes for selection that are undesirable for interpretability.[6] Examples of LISA can be found in Figure 1 and Table 3. With 768 dimensions,

producing a single LISA vector would require 768 inferences of SFAM, a computationally expensive operation. To address this, we produce the LISA representations for 1,000,000 random Reddit posts from MUD. We then distill into a new EncT5 model with 768 regression labels. We hold-out 10,000 examples as a validation set. After distillation to the dedicated model, the 768-dimensional style vector can be produced in a single forward pass with minimal degradation (validation MSE = 0.005).

### 3.3 LISA Style Embeddings

We experiment with two different simple and interpretable embedding layers, a weight vector ($w_{768}$) and a weight matrix ($W_{768 \times 64}$). We attach these on top of the LISA model and train just the layer using a contrastive learning objective and triplet loss (Khosla et al., 2020; Schroff et al., 2015). We also experiment with two different authorship datasets

---

[6]We reject style attributes that are $> 120$ characters, are not pure ASCII, contain "not" or "avoids" (negative statements), or contain quotes or the word "mentions" (these attributes tend to be more relevant to content than style).

| Model | Formal | Complex | Numb3r | C'tion | Avg | Interpretable |
|---|---|---|---|---|---|---|
| Random Baseline | 0.50/0.50 | 0.50/0.50 | 0.50/0.50 | 0.50/0.50 | 0.50/0.50 | |
| *Content-Aware Representations* | | | | | | |
| SBERT | 0.78/0.00 | 0.54/0.01 | 0.81/0.04 | 0.86/0.00 | 0.75/0.01 | ✗ |
| LUAR | 0.80/0.14 | **0.67**/0.00 | 0.74/0.03 | 0.77/0.00 | 0.75/0.04 | ✗ |
| *Content-Independent Style Representations* | | | | | | |
| LIWC | 0.52/ - | 0.52/ - | 0.50/ - | **0.99**/ - | 0.63/ - | ✓ |
| Wegmann et al. (2022) | **0.84/0.69** | 0.59/**0.26** | 0.56/0.03 | 0.96/**0.02** | 0.74/**0.25** | ✗ |
| LISA | 0.69/0.07 | 0.57/0.01 | 0.80/0.03 | 0.77/0.00 | 0.71/0.03 | ✓ |
| LISA (Wegmann + $w$) | 0.72/0.07 | 0.61/0.03 | 0.81/**0.08** | 0.68/0.00 | 0.71/0.05 | ✓ |
| LISA (Wegmann + $W$) | 0.66/0.03 | 0.56/0.01 | 0.70/0.01 | 0.87/0.00 | 0.70/0.01 | ✓ |
| LISA (LUAR + $w$) | 0.73/0.05 | 0.65/0.00 | **0.85**/0.03 | 0.92/0.00 | **0.79**/0.02 | ✓ |
| LISA (LUAR + $W$) | 0.81/0.07 | 0.56/0.01 | 0.74/0.03 | 0.82/0.00 | 0.73/0.03 | ✓ |

Table 4: Accuracy scores on STEL/STEL-or-Content, an evaluation framework for style measures proposed by Wegmann and Nguyen (2021) and Wegmann et al. (2022). "LIWC" results are from Wegmann and Nguyen (2021). "LISA" is the 768-dimensional style vector. "LISA (...)" uses LISA embeddings with the training dataset and embedding layer type denoted in (...). Gray indicates worse than random baseline performance on the adversarially challenging STEL-or-Content task. All approaches underperform on STEL-or-Content, but LISA approaches outperform or closely match existing style representation choices on STEL, while providing interpretability.

from prior works to train the embedding layer; we refer to these datasets as the Wegmann dataset (Wegmann et al., 2022) and the LUAR dataset (Rivera-Soto et al., 2021). Like the prior work, we assume an author has consistent style between their different texts. Given some anchor text by an author, we use another text by the same author as a positive example, and text by a different author as a negative example for our triplets. This objective minimizes the distance between two texts by the same author and maximizes the distance between texts by different authors, learning a meaningful metric.

## 4 Results

We first evaluate to what degree SFAM, which is ultimately used to build LISA representations, learns useful stylometric annotation capabilities that align with human reviewers. We then evaluate LISA itself on STEL, a framework purpose-built for evaluating the quality of style measures (Wegmann and Nguyen, 2021). All evaluations were completed after the collection of the STYLEGENOME dataset.

**Correlation to Human Judgments** We conduct a broad set of 55 studies across 21 datasets in 7 distinct categories of linguistic style and authorship dimensions in Table 2. We measure the correlation of SFAM's agreement scores to human judgments. SFAM performs stronger on dimensions like formality, sentiment, and emotion than dimensions like linguistic acceptability. This is likely an artifact

of the effectiveness of GPT-3 in annotating these categories, an expected result given prior work has shown language models struggle with identifying these features (Warstadt et al., 2020). Interestingly, SFAM demonstrates some limited ability to perform authorship profiling, a task adjacent to stylometry. The ability to probe SFAM in an interpretable manner helps identify which categories of features it can reliably represent, whereas prior approaches were more opaque. Overall, the Table 2 results demonstrate SFAM's annotations do correlate with human judgments on some important dimensions of style. We hypothesize future research with larger datasets (> 10,000 posts), more diverse sources of texts, and larger and more performant LLMs may further broaden and improve learned stylometric annotation capabilities.

**STEL** In Table 4, we provide the results of evaluating LISA using STEL. The STEL task evaluates whether two texts with similar styles can be matched using the distance/similarity metric defined by a style representation. We compare with other content-independent style representations, or methods that explicitly limit representation of content in favor of style. LISA explicitly limits the representation of content through the 768 style-focused attributes that act as a bottleneck. Content-aware representations like SBERT, on the other hand, have direct access to the text and may be able to represent the content in the text to an extreme degree, representing the usage of a specific rare

| Text | Style Attribute |
|------|-----------------|
| Subscribed. Interesting idea. I would like to see some advanced stats on hitting percentages to different locations on the court. For example, having the court broken up into maybe 12 zones and then hitting percentages from each position to those zones. I remember seeing an article that did this years ago and I have never been able to find anything online. I said recently on this sub that the deep angle shot from the left side or right side was the highest percentage shot in volleyball, but I was not able to back up my claim with any sources or anything. Anyways, I am a VB nerd, no doubt. Interested to see what direction you take this. Cheers! | The author is being polite. |
| Yeah I also work in QA, and seeing this kind of stuff get released is maddening. About a year ago working on a new platform we were seeing bugs in the hundreds each week, we pushed back the release of the product 3 months because basically it didn't work. If it was up to the devs, they'd have released it on time, because the stuff they'd written code for worked. Thorough doesn't even cover the work we go through every 3 months, and Niantic's approach seems completely amateur from this side. They're putting bandaids on problems and hiding things like the 3 step problem behind curtains without seemingly fixing anything, although I do have to say their balance tweaks to battling have been a big step in the right direction. | The author is using a personal anecdote to illustrate their point. |
| Thank you. I'd be interested in reading more about your experiences, in addition to the "American Wedding" story. Are you watching the stream? I wish there was a way to find out how many people in the world are watching it. The music is lovely, huh? God damn. He's got his bunny Fair Isle sweater on, drinking Dunkin' Donuts coffee. I would have thought him a Starbucks man. :-) | The author is using an emoji. |

Table 5: Sentence-level LISA vectors over each sentence from a longer passage of text can help identify and quantify which sentences contribute to overall style attributes scored on the longer passage providing granular interpretability.

word or discussion of a specific concept. We provide the results of content-aware representations simply for reference. We find LISA embeddings are able to closely match (and on average slightly outperform) prior style representations on STEL while providing interpretability.

**Sentence-Level Interpretability** In Table 5, we demonstrate how visualizing a dimension of sentence-level LISA vectors can help explain which sentences contribute to a dimension activated on a passage-level LISA vector.

**Forensic Interpretability** Typically for authorship attribution tasks, content-aware representations that capture both content and style are used to make a determination. Author style, however, is still an important component in determining attribution (Rivera-Soto et al., 2021). Offering a clear explanation and presenting supporting evidence is crucial, particularly in the context of forensic analysis, such as when presenting evidence in a court trial. Explainability has often been overlooked in neural approaches to authorship attribution tasks. To motivate this as a future research direction using our interpretable stylometric representations and our general approach, we provide an example of explanations on the Contrastive Authorship Verifi-

cation task from Wegmann et al. (2022) in Table 6 with LISA (LUAR + $W$). Further examples and discussion on how the top common and distinct style attributes are ranked can be found in Appendix G.

### 4.1 Error Analysis

We highlight insights and observations around common failure modes of our technique in this section. We annotate the common failure modes with their percentage rate of occurrence.[7]

**Content vs. Style Attributes (3%)** It is unclear whether style and content can truly be separated as some content features are important for style or profiling an author (Jafaritazehjani et al., 2020; Bischoff et al., 2020; Patel et al., 2022). Even after filtering, 3% of dimensions of LISA still represent content. For example, "The author is using words related to the game they are discussing". However, while LISA may have the ability to represent that two texts are both discussing the topic of video games, it does not have the direct ability a content-aware approach would of representing which specific video game is being discussed, due to the

---

[7]We manually inspect a small sample set of 2,000 style attribute annotations (the top 20 style attributes for 100 random texts) by LISA.

| Texts | Top 3 Common/Distinct Style Attributes |
|---|---|
| **Anchor:** Devices that use two pronged instead of three pronged plugs are required to meet certain safe design requirements. Among other things, if a device has a switch, the switched line MUST BE hot, not neutral. The polarized plugs make sure that the right prong/wire is hot. This is why devices that have no switches (primarily wall warts) need not have polarized plugs. **Same Author:** Your diaphragm would be trying to contract against the air pressure in your lungs. That's why deep sea diving requires regulators, to match the pressure of the air supply to the pressure surrounding your rib cage. You can breathe against a maximum of about 1/2 PSI, which is not enough pressure to adequately oxygenate your blood. | (0.89, 1.00) − The author is using a scientific approach. (0.96, 0.98) − The author is using a combination of technical terms and everyday language. (0.91, 0.84) − The author is using formal and professional language. |
| **Different Author:** That's great! I'm glad it seems to be finding its' niche. Now if they could just make a Star Wars version of this game, I'd happily swallow that fat learning curve and overcome my frustrations with the combat system. ;) | (0.06, 0.99) − The author is using words related to the game they are discussing. ∗ (0.00, 0.88) − The author is using an emoji. (0.02, 0.87) − The author uses an emoticon at the end. |

Table 6: Example interpretable explanations on the Contrastive Authorship Verification task. The top style attributes in common between the Anchor text and a text by the Same Author are shown. The top distinct style attributes between the Anchor text and a text by a Different Author are also shown. The scores of each style attribute against the texts is shown in (•, •/•). Attributes annotated with ∗ blur the line between style and content. Error analysis can be found in Section 4.1. Further examples and details on style attribute ranking can be found in Appendix G.

limited set of 768 features that act as a bottleneck. Our approach also allows visibility into understanding how much of the representation derives from content-related features, while other neural representations are opaque and may use content-related features in a way that cannot be easily assessed.

**Conflating Style Attributes with Content (2%)** For some style attributes, LISA conflates the content of text with the presence of the style attribute. For example, "The author is cautious", may have a high agreement score on any text containing the word "caution" even if the author is not actually expressing caution in the text.

**Spurious Correlations (6%)** For other style attributes, LISA has learned spurious correlations. For example, "The author uses two exclamation marks", often has a high agreement score on any text that is exclamatory in nature, but does not actually use exclamation marks. An example can be found in Table 3.

**Fundamental Errors (10%)** LISA sometimes produces a high agreement score for text displaying the polar opposite of a style attribute or produces a high agreement score for an attribute that simply is not present in the text. Table 3 demonstrates some of these incorrect examples. Inspecting our dataset, this error happens both due to EncT5's internal representations likely aligning on relatedness instead of similarity (Hill et al., 2015) and due to

hallucination and annotation errors by GPT-3. Hallucinated generations is a common issue with any LLM-guided approach and we discuss it further in Limitations along with potential future mitigations.

## 5 Conclusion

In this work, we propose a promising novel approach to learning interpretable style representations. To overcome a lack of stylometrically annotated training data, we use a LLM to generate STYLEGENOME, a synthetic stylometry dataset. Our approach distills the stylometric knowledge from STYLEGENOME into two models, SFAM and LISA. We find that these models learn style representations that match the performance of recent direct neural approaches and introduce interpretability grounded in explanations that correlate with human judgments. Our approach builds towards a research direction focused on making style representations more useful for downstream applications where such properties are desirable such as in a forensic analysis context. Future directions that introduce human-in-the-loop supervised annotations or newer, larger, and better aligned LLMs for annotation have the potential to yield further gains in both performance and interpretability.

**Model and Data Release** We release our dataset (STYLEGENOME) and our two models (SFAM and LISA) to further research in author style.

## Limitations and Broader Impacts

**Limitations** Handcrafted features by forensic linguists typically rely on frequency counts of word usage, usage of unique words or phrases, etc. (Mosteller and Wallace, 1963). The space of these kinds of features is non-enumerable and would not be well-represented with our technique that scores a fixed set of 768 interpretable features. Pure neural approaches may capture these kinds of features, but are non-interpretable and may capture undesirable content-related features. We explicitly trade-off the use of these kinds of features in this work to achieve interpretability. While we demonstrate our synthetic annotations are enough for a model to learn to identify stylistic properties in text in Table 2, they cannot be fully relied on yet for the reasons we discuss in Section 4.1. As large language models scale and improve, however, we believe this work could benefit from increasing coherency and decreasing hallucination in the annotations (Kaplan et al., 2020). STYLEGENOME is collected only on 10,000 English Reddit posts, however, larger datasets may improve performance as we show in Figure 2 and future research in multilingual LLMs may make it feasible to replicate this procedure for other languages.

**Ethical considerations** Style representations are useful for text style transfer (Riley et al., 2021) and in manipulating the output of machine generated text to match a user's style, for example, in machine translation (Niu et al., 2017; Rabinovich et al., 2017). While style transfer can be a useful benign commercial application of this work, superior style representations may aid the impersonation of authors. We demonstrate how style representations may aid legitimate cases of authorship attribution, a task that is typically done by forensic linguist experts. Our work introduces an interpretable approach, an important step in legitimizing the use of computational models for authorship attribution by providing explanations for predictions that can be audited and verified.

**Diversity and inclusion** We believe style representations that capture wider dimensions of style can help aid in analyzing and representing minority writing styles in downstream applications like style transfer.

## Acknowledgements

This research is based upon work supported in part by the DARPA KAIROS Program (contract FA8750-19-2-1004), the DARPA LwLL Program (contract FA8750-19-2-0201), the IARPA HIATUS Program (contract 2022-22072200005), and the NSF (Award 1928631). Approved for Public Release, Distribution Unlimited. The views and conclusions contained herein are those of the authors and should not be interpreted as necessarily representing the official policies, either expressed or implied, of DARPA, IARPA, NSF, or the U.S. Government.

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

# A Prompt Templates

## A.1 Open-ended Prompt Templates

- **Grammar Style**: `Write a long paragraph describing the unique grammar style of the following passage without referring to specifics about the topic.`

    `Passage: {{passage}}`

    `Description:`

- **Vocabulary Style**: `Write a long paragraph describing the unique vocabulary style of the following passage without referring to specifics about the topic.`

    `Passage: {{passage}}`

    `Description:`

- **Punctuation Style**: `Write a long paragraph describing the unique punctuation style of the following passage without referring to specifics about the topic.`

    `Passage: {{passage}}`

    `Description:`

- **Grammar Errors**: `Write a long paragraph describing the grammar errors (if any) of the following passage without referring to specifics about the topic.`

    `Passage: {{passage}}`

    `Description:`

- **Spelling Errors**: `Write a long paragraph describing the spelling errors (if any) of the following passage without referring to specifics about the topic.`

    `Passage: {{passage}}`

    `Description:`

- **Forensic Linguist**: `Write a long paragraph describing the unique stylometric features of the following passage without referring to specifics about the topic from the perspective of a forensic linguist psychoanalyzing the writer.`

    `Passage: {{passage}}`

    `Description:`

## A.2 Targeted Prompt Templates

We use the following template:

```
{{target_feature_definition}}

Write a description of whether the author of the following passage has any
{{target_feature}}?

Passage:  {{passage}}

Description:
```

for targeted prompts, substituting `{{target_feature}}` with each of the following targeted features:

- figurative language
- sarcasm
- sentence fragment
- run on sentences
- an active voice
- a passive voice
- agreement errors
- male pronouns
- female pronouns
- prosocial behaviors
- antisocial behaviors
- being polite
- showing interpersonal conflict
- moralizing
- communication words
- indicators of power
- talk of achievement
- indication of certitude
- being tentative
- insight
- all or none thinking
- words related to memory
- positive emotion
- negative emotion
- anxiety
- anger
- sadness
- swear words
- positive tone

- negative tone
- neutral tone
- words related to auditory perception
- words related to visual perception
- words related to space perception
- words related to motion perception
- words related to attention
- words related to allure
- words related to curiosity
- words related to risk
- words related to reward
- words expressing needs
- words expressing wants
- words expressing acquisition
- words expressing lack
- words expressing fulfillment
- words expressing fatigue
- words expressing illness
- words expressing wellness
- words related to mental health
- words related to food or eating
- words related to death
- words related to self-harm
- sexual content
- words related to leisure
- words related to home
- words related to work
- words related to money
- words related to religion

- words related to politics
- words related to culture
- swear words
- foreign words
- scholarly words
- slang words
- social media slang words
- filler words
- words focusing on the past
- words focusing on the present
- words focusing on the future
- words related to time
- misspelled words
- repeated words
- words expressing quantity
- words indicating family
- words indicating friends
- words indicating men
- words indicating women
- words indicating pets
- words indicating social status
- words indicating poverty
- words indicating wealth
- punctuation symbols
- hyphenated words
- oxford comma
- parentheticals
- numbers
- elongated words

To give GPT-3 more context, we also substitute `{{target_feature_definition}}` with a definition of the target feature, also generated by GPT-3. The full set of targeted prompts can be found in the released source package for this paper.

### A.3 Standardization Prompt Templates

The descriptions of style generated from the prompts in Appendix A.1 and Appendix A.2 are substituted into the following standardization prompt:

```
Here's a description of an author's writing style for a passage:  {{description}}

Rewrite this description as a long list of short sentences describing the author's
writing style where each sentence is in the format of "The author is X." or "The
author uses X.".

Output:
```

which transforms the verbose descriptions into short, declarative, uniform sentences beginning with "The author...," which are the final style attributes used in building the STYLEGENOME dataset that SFAM is trained on.

## B  Effect of STYLEGENOME Dataset Size

When training SFAM, we experiment with artificially limiting the size of the synthetic dataset, by limiting the number of authors in the dataset, to determine the effect of dataset size on the validation performance. In Figure 2, we find that as the synthetic dataset grows, validation performance improves and SFAM generalizes to better predict agreement scores for unseen style attribute and text pairs.

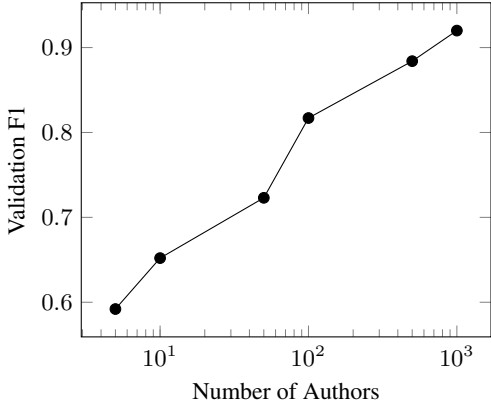

Figure 2: Best F1 achieved by SFAM on a held-out validation set of examples at various dataset sizes.

## C  Annotation Prompts Ablation

| Prompts Used to Generate STYLEGENOME | Validation F1 |
|---|---|
| Open-ended Prompts | 0.865 |
| Targeted Prompts | 0.898 |
| Open-ended Prompts & Targeted Prompts | **0.920** |

Table 7: Best F1 achieved by SFAM on a held-out validation set of examples with different sets of Stage 1 prompts used to annotate Reddit posts and generate the synthetic training data used during distillation.

## D  STYLEGENOME Annotation Cost

Our inference cost with the OpenAI API was priced at $0.02 / 1K tokens, with a cost of ~$8 to annotate 10 Reddit posts by a single author with all of our prompts. Our full dataset of 1,000 authors cost ~$8,000 to annotate.

# E  Training Details

## E.1  SFAM

We use the EncT5 architecture (Liu et al., 2021) with a binary classifier in Hugging Face (Wolf et al., 2019). We randomly sample training batches of size 1,440 and use the AdamW optimizer with a learning rate of 0.001. We employ early stopping with a threshold of 0.01 on the validation set F1 metric and a patience of 50 batches.

## E.2  LISA

We use the EncT5 architecture (Liu et al., 2021) with 768 regression labels in Hugging Face (Wolf et al., 2019) and use `MSELoss`. We randomly sample training batches of size 1,440 and use the AdamW optimizer with a learning rate of 0.001. We employ early stopping with a threshold of 1e-6 on the validation set MSE metric and a patience of 20 epochs.

## E.3  LISA Embedding Layers

We experiment with two types of embedding layers $w$ and $W$. We also experiment with two training datasets, the Wegmann dataset (Wegmann et al., 2022) and the LUAR dataset (Rivera-Soto et al., 2021). For LUAR, we use the train split with 5% held out as validation and we sample random authors as negative examples. For Wegmann, we use the Conversation dataset train split and the dev split as validation. We use a margin of 1.0, a batch size of 32, an AdamW optimizer with a learning rate of 0.001, and employ early stopping with a threshold of 0.001 on the validation set loss.

# F  Full SFAM Evaluation Results

| Type | Dataset | Style Attribute | Spearman Correlation ($\rho$) |
|---|---|---|---|
| Formality | Formality in Online Communication | The author uses informal language. | 0.599 |
| | Grammarly's Yahoo Answers Formality Corpus | | 0.200 |
| Sentiment | Yelp Reviews | The author uses a negative tone. | 0.788 |
| | IMDB Large Movie Review Dataset | | 0.665 |
| | Amazon Customer Reviews Dataset | | 0.432 |
| | Rotten Tomatoes Movie Review Data | | 0.463 |
| | App Reviews | | 0.350 |
| | Twitter Sentiment Analysis Training Corpus | | 0.299 |
| Emotion | DAIR.AI Emotion (Love vs. Anger) | The author is expressing {{emotion}}. | 0.542 |
| | DAIR.AI Emotion (Joy vs. Sad) | | 0.531 |
| | GoEmotions (Love vs. Anger) | | 0.769 |
| | GoEmotions (Joy vs. Sadness) | | 0.639 |
| | GoEmotions (Disgust vs. Desire) | | 0.630 |
| | GoEmotions (Disappointment vs. Admiration) | | 0.571 |
| | GoEmotions (Pride vs. Embarassment) | | 0.419 |
| | GoEmotions (Nervousness vs. Optimism) | | 0.447 |
| | GoEmotions (Disapproval vs. Approval) | | 0.432 |
| | GoEmotions (Admiration vs. Neutral) | | 0.578 |
| | GoEmotions (Amusement vs. Neutral) | | 0.508 |
| | GoEmotions (Anger vs. Neutral) | | 0.372 |
| | GoEmotions (Annoyance vs. Neutral) | | 0.355 |
| | GoEmotions (Approval vs. Neutral) | | 0.257 |
| | GoEmotions (Caring vs. Neutral) | | 0.303 |
| | GoEmotions (Confusion vs. Neutral) | | 0.343 |
| | GoEmotions (Curiosity vs. Neutral) | | 0.464 |
| | GoEmotions (Desire vs. Neutral) | | 0.315 |
| | GoEmotions (Disappointment vs. Neutral) | | 0.317 |
| | GoEmotions (Disapproval vs. Neutral) | | 0.284 |
| | GoEmotions (Disgust vs. Neutral) | | 0.307 |
| | GoEmotions (Embarrassment vs. Neutral) | | 0.173 |
| | GoEmotions (Excitement vs. Neutral) | | 0.295 |
| | GoEmotions (Fear vs. Neutral) | | 0.309 |
| | GoEmotions (Gratitude vs. Neutral) | | 0.626 |
| | GoEmotions (Grief vs. Neutral) | | 0.093 |
| | GoEmotions (Joy vs. Neutral) | | 0.390 |
| | GoEmotions (Love vs. Neutral) | | 0.497 |
| | GoEmotions (Nervousness vs. Neutral) | | 0.171 |
| | GoEmotions (Optimism vs. Neutral) | | 0.407 |
| | GoEmotions (Pride vs. Neutral) | | 0.097 |
| | GoEmotions (Realization vs. Neutral) | | 0.121 |
| | GoEmotions (Relief vs. Neutral) | | 0.123 |
| | GoEmotions (Remorse vs. Neutral) | | 0.276 |
| | GoEmotions (Sadness vs. Neutral) | | 0.413 |
| | GoEmotions (Surprise vs. Neutral) | | 0.317 |
| Author Profiling | Political Slant | The author is a Democrat. | 0.005 |
| | Twitter User Gender Classification | The author is female. | 0.166 |
| | African-American Vernacular English | The author uses African-American Vernacular English. | 0.238 |
| | Shakespeare (Early Modern English) | The author uses Early Modern English. | 0.108 |
| | Wikipedia Bias | The author has a biased point of view. | 0.014 |
| Harmful Speech | HateSpeech18 | The author's writing contains hate speech. | 0.229 |
| | Offensive Social Media | The author uses offensive language. | 0.401 |
| Text Simplification | Simple Wikipedia | The author uses simple language. | 0.043 |
| | ASSET | | 0.050 |
| Linguistic Acceptability | CoLA | The author uses incorrect grammar. | 0.078 |
| | BLiMP | | 0.020 |
| **Average** | | | **0.342** |

Table 8: Correlation of agreement scores produced by SFAM against human judgments on texts over a wide variety linguistic and authorship dimensions. The natural language style attributes used as input to SFAM when producing the agreement scores for each dataset are also provided.

# G    Interpretable Authorship Verification

| Texts | Top 3 Common/Distinct Style Attributes |
|---|---|
| **Anchor:** Devices that use two pronged instead of three pronged plugs are required to meet certain safe design requirements. Among other things, if a device has a switch, the switched line MUST BE hot, not neutral. The polarized plugs make sure that the right prong/wire is hot.
This is why devices that have no switches (primarily wall warts) need not have polarized plugs.
**Same Author:** Your diaphragm would be trying to contract against the air pressure in your lungs. That's why deep sea diving requires regulators, to match the pressure of the air supply to the pressure surrounding your rib cage. You can breathe against a maximum of about 1/2 PSI, which is not enough pressure to adequately oxygenate your blood. | (0.89, 1.00) − The author is using a scientific approach.
(0.96, 0.98) − The author is using a combination of technical terms and everyday language.
(0.91, 0.84) − The author is using formal and professional language. |
| **Different Author:** That's great! I'm glad it seems to be finding its' niche. Now if they could just make a Star Wars version of this game, I'd happily swallow that fat learning curve and overcome my frustrations with the combat system. ;) | (0.06, 0.99) − The author is using words related to the game they are discussing. ∗
(0.00, 0.88) − The author is using an emoji.
(0.02, 0.87) − The author uses an emoticon at the end. |
| **Anchor:** Not sure what the income tax is in Germany, but in the Netherlands the income can be up 50% for the higher income classes.
**Same Author:** The salaries in the US alway blow my mind. A software developer in Amsterdam gets like €40.000/year, maybe €50.000/year if your good, and maybe €60.000/year if you're some kind of manager. Anything position over €100.000/year is basically running the entire company. | (0.84, 0.90) − The author is using words indicating poverty.
(0.87, 1.00) − The author is using words indicating wealth.
(0.80, 0.93) − The author is using words related to money. |
| **Different Author:** How would you even test this software? The setup would be just insane. | (0.00, 1.00) − The author is comfortable with technology.
(0.00, 0.85) − The author is discussing a product. ∗
(0.13, 0.92) − The author is using formal and professional language. |
| **Anchor:** If only there was something he could have done to avoid this backlash. Like maybe not acting like a complete d**khead.
**Same Author:** I take issue with a faster landing being marked as less skilled. By that logic the slowest, smoothest possible landing would be the most skilled and that is plain wrong. Maybe war machine intentionally does faster and harder landings. | (1.00, 0.96) − The author is emphasizing the contrast between the two ideas.
(0.78, 0.89) − The author is able to draw conclusions.
(0.97, 0.86) − The author is using an all-or-none thinking style. |
| **Different Author:** She was the Ronald Reagan of the UK in the same time period. | (0.83, 0.05) − The author is describing sexual content. ∗
(0.38, 0.94) − The author is using words related to politics. ∗
(0.74, 0.38) − The author is using parentheticals. |

Table 9: Example interpretable explanations on the Contrastive Authorship Verification task. The top style attributes in common between the Anchor text and a text by the Same Author are shown. The top distinct style attributes between the Anchor text and a text by a Different Author are also shown. The scores of each style attribute against the texts is shown in (•, •/•). We manually inspect the style attributes and annotate them as reasonable, plausible, or incorrect explanations. Attributes annotated with ∗ blur the line between style and content. Error analysis can be found in Section 4.1.

We perform this task with LISA (LUAR + $W$) and demonstrate interpretability on a few task instances. To rank the top common or distinct style attributes between two style vectors $\vec{v_1}$ and $\vec{v_2}$, we perform a simple calculation. We first calculate the contribution of each dimension $d$ to the Euclidean distance as a measure of the general importance of each dimension. The importance score is defined as:

$$\mathcal{I}(d) = \|\vec{v_2} - \vec{v_1}\|_2 - \sqrt{\sum_{k=0}^{D} \begin{cases} (\vec{v_{2k}} - \vec{v_{1k}})^2 & k \neq d \\ 0 & k = d \end{cases}}$$

To retrieve the top common style attributes, we rank the dimensions, and the corresponding style attributes, in descending order by the following score function:

$$\text{SCORE}_{\text{common}}(d) = \frac{\mathcal{I}(d)}{\sum_{k=0}^{D} \mathcal{I}(k)} * \vec{v_1}_d * \vec{v_2}_d$$

To retrieve the top distinct style attributes, we rank the dimensions, and the corresponding style attributes, in descending order by the following score function:

$$\text{SCORE}_{\text{distinct}}(d) = \frac{\mathcal{I}(d)}{\sum_{k=0}^{D} \mathcal{I}(k)} * \max(\vec{v_1}_d, \vec{v_2}_d) * \left(1.0 - \min(\vec{v_1}_d, \vec{v_2}_d)\right)$$

## H   Resources

We provide links and citations to resources used in this paper which provide license information, documentation, and their intended use. Our usage follows the intended usage of all resources.

We utilize the following models:

- GPT-3$_{175B}$ (`text-davinci-003`) (Brown et al., 2020; Ouyang et al., 2022)

- EncT5 (`t5-base`) (Devlin et al., 2019; Liu et al., 2019)

- DistilRoBERTa (`nli-distilroberta-base-v2`) (Sanh et al., 2019; Liu et al., 2019; Devlin et al., 2019; Reimers and Gurevych, 2019)

- Learning Universal Authorship Representations (LUAR) Embedding model (Rivera-Soto et al., 2021)

- Style embedding model from Wegmann et al. (2022)

We utilize the following datasets:

- Reddit Million User Dataset (Khan et al., 2021; Andrews and Bishop, 2019)

- STEL dataset (Wegmann and Nguyen, 2021)

- Contrastive Authorship Verification dataset (Wegmann et al., 2022)

- Formality in Online Communication (Pavlick and Tetreault, 2016)

- Grammarly's Yahoo Answers Formality Corpus (Rao and Tetreault, 2018)

- Yelp Reviews Dataset (Zhang et al., 2015)

- IMDB Large Movie Review Dataset (Maas et al., 2011)

- Amazon Customer Reviews Dataset (Amazon.com, 2018) – `https://s3.amazonaws.com/amazon-reviews-pds/readme.html`

- Rotten Tomatoes Movie Review Data (Pang and Lee, 2005)

- App Reviews Dataset (Grano et al., 2017)

- Twitter Sentiment Analysis Training Corpus (Naji, 2012)

- DAIR.AI Emotion Dataset (Saravia et al., 2018)

- GoEmotions Dataset (Demszky et al., 2020)

- Political Slant Dataset (Prabhumoye et al., 2018)

- Twitter User Gender Classification Dataset (CrowdFlower, 2017) – `https://www.kaggle.com/datasets/crowdflower/twitter-user-gender-classification`

- African-American Vernacular English Dataset (Groenwold et al., 2020)

- Shakespeare Dataset (Xu, 2017)

- Wikipedia Bias Dataset (Pryzant et al., 2020)

- HateSpeech18 (de Gibert et al., 2018)

- Offensive Social Media Dataset (Atwell et al., 2022)

- Simple Wikipedia Dataset (Coster and Kauchak, 2011)

- ASSET (Alva-Manchego et al., 2020)

- CoLA (Warstadt et al., 2019)

- BLiMP (Warstadt et al., 2019)

We utilize the following software:

- Transformers ([Wolf et al., 2019](#))

- Sentence-Transformers ([Reimers and Gurevych, 2019](#))

- `emoji` – `https://pypi.org/project/sentence-splitter/`

- `sentence-splitter` – `https://pypi.org/project/sentence-splitter/`

We estimate the total compute budget and detail computing infrastructure used to run the computational experiments found in this paper below:

- 1x NVIDIA RTX A6000 / 30GB RAM / 4x CPU – 230 hours