# OpenReview forum: "Learning Interpretable Style Embeddings via Prompting LLMs"
_EMNLP/2023/Conference — EMNLP 2023 Findings_

### Official Review · Reviewer_cLTM · 2023-08-04

**Soundness:** 3

**Excitement:**

3: Ambivalent: It has merits (e.g., it reports state-of-the-art results, the idea is nice), but there are key weaknesses (e.g., it describes incremental work), and it can significantly benefit from another round of revision. However, I won't object to accepting it if my co-reviewers champion it.

**Paper Topic And Main Contributions:**

This paper describes a data augmentation framework to learn interpretable style embeddings. To create synthetic training data, GPT-3 is prompted to produce natural language descriptions of the writing style of a given document, using a hand-crafted set of 87 prompts based on a survey of stylometry literature. An additional 6 open-ended prompts aim to elicit a general description of the writing style. The resulting natural language descriptions are post-processed into separate clauses each starting with "The author [...]" (with *yet another* LLM prompt), each of which is treated as a separate interpretable stylistic attribute. The resulting synthetic dataset is called "StyleGenome."

To make use of StyleGenome, "LISA Style Vectors" are created based on an arbitrary choice of D = 768 dimensions. The first 87 dimensions correspond to the features of targeted prompts. The remaining 681 are downselected from the ~1.3M style attributes using filtering heuristics: appear for at least 10 authors but no more than 600 authors. Also, once an attribute is selected, duplicates are avoided using SBERT cosine similarity. Also, attributes are rejected if they are too long, contain specifics like "mentions", and negations. Finally, an EncT5 model is trained with a binary classifier head to produce predictions efficiently on held-out data, where natural language descriptions allow for generalization across various natural language stylistic descriptors.

The resulting features are evaluated using an evaluation framework from Wegmann and Nguyen (2021) which aims to characterize style  discrimination ability for categories such as formal and complex. The proposed LISA embeddings are found to be competitive with recent alternatives while having superior interpretability.

**Reasons To Accept:**

* Creative use of LLMs to produce a new dataset with style attribute annotations.
* The distillation framework appears to successful transfer knowledge from GPT-3 into a significantly smaller / more efficient model.
* Evaluation comparing to both human judgements as well as other recent style embedding approaches, including error analysis
* Release of dataset and models

**Reasons To Reject:**

Main concerns:
* I would have liked to see more investigation of the quality of the StyleGenome dataset. For example, although Appendix B provides some evidence that there's signal there, the trend could be more complicated when looking at generalization for individual attributes, and there is the risk that essentially noise/hallucinations from the LLM are being fit.

* It's unclear what feedback was used to guide the various manual decisions that went in to the pipeline (e.g., prompt design, normalization, filtering). What this all done on the basis of some validation data or manual inspection?

* It's unclear how distinct the various stylistic attributes are since they're based on natural language descriptions, and SBERT similarity won't identify all cases of of near duplicate attributes. Perhaps some kind of dimensionality reduction could help.

Minor concerns:

* For selecting the attributes to keep, is there a more standard / less heuristic approach, such as TFIDF, that could be used?
* I found it pretty difficult to wrap my head around the mapping from an open-set of natural language attributes to a fixed and closed set of features. The presentation could likely be improved to make this easier.
* The paper cited for the "Million User Dataset" does not appear to describe this dataset.
* I would have liked to see other versions of StyleGenome compared for different LLMs, especially open-source ones.

**Reproducibility:**

4: Could mostly reproduce the results, but there may be some variation because of sample variance or minor variations in their interpretation of the protocol or method.

**Reviewer Confidence:**

5: Positive that my evaluation is correct. I read the paper very carefully and I am very familiar with related work.

---

> ### Author Rebuttal · Authors · 2023-08-23
>
> We thank the reviewer for their comments.
>
> We respond to some of the main and minor concerns below and believe they will be adequately addressed in the camera-ready. We believe they are fully addressed here as well, but if they are not fully addressed, we are happy to respond further:
>
> 1. **StyleGenome:** We don't directly analyze StyleGenome as it is very large (millions of annotations by GPT-3). As you mentioned, however, we provide some evidence the dataset provides signal by evaluating a downstream model trained using the dataset. By design, this is a synthetic dataset, so we don't claim it is 100% correct and clearly state the limitations of an unsupervised approach (Line 430-431) and provide error analysis, however, we release it following other papers that have recently released synthetic datasets for reproducibility (in case others want to train models similar to our SFAM/LISA models using this dataset). It was also fairly expensive to produce (~$8,000 - Line 950), so we believe it could be a useful reusable resource to the community that is lacking these kinds of resources around text style annotations.
>
> 2. **Pipeline Decisions:** Many of these decisions such as prompt design were manually made at the time of producing the dataset. Decisions like filtering were simply made to get to an embedding of D=768 dimensions (an arbitrary choice as we state, but is the traditional choice of the # of dimensions with BERT models). Evaluations on SFAM / LISA were performed only at the end after training. We can make a small revision in our camera-ready to make these minor manual choices more clear.
>
> 3. **Near-Duplicate Dimensions:** That's correct, there may be some near-duplicates in the dimensions. However, the main reason we filter out near-duplicates and less discriminative features is to reduce our feature space from ~ 1.3M style attributes to a more manageable 768 dimensions, since a 1.3M dimensional vector would be too unwieldy. Further, human de-duplication is difficult as it requires approx. 768 choose 2 (~300K) pairwise inspections, so we rely solely on the automated near-duplicate detection using SBERT. In the camera-ready draft, however, we can mention the fact that a small number of near-duplicates may remain even after SBERT-filtering.
>
> We also briefly respond to your minor concerns:
>
> 1. **TF-IDF**: We do actually use a TF-IDF library to this, this is evident in our code package we submitted, and we treat "style attributes = terms" and "authors = documents" using it to select attributes that are most discriminative between authors similar to how TF-IDF will upweight terms most important to a document. We allude to this in the paper when we discuss selecting discriminative style attributes by using how often they appear for each author (or "document frequency"), but we can make this more clear in the camera-ready along with our code package.
>
> 2. **Presentation of Features:** Thanks for the suggestion, that's helpful, we'll definitely take this into consideration when rephrasing for camera-ready.
>
> 3. **Million User Dataset citation:** You're correct, our citation is only half-correct. Thank you for pointing this out. Recent guidance from the authors of that dataset after we submitted this draft states the full citation should be two citations: (Andrews and Bishop 2019; [Khan et al. 2021](https://arxiv.org/pdf/2105.07263.pdf)). We will update this citation in the camera-ready as well.
>
> 4.  **Open-source LLMs:** We briefly experimented with open source LLMs for this task early on in our research, however, at the time this research began, many of the current instruction-tuned models (LLaMa-2-chat) were not available. It's possible they are also performant for this task, however, we release our synthetic dataset (StyleGenome) and our own distilled, efficient, smaller open source models based on T5 (SFAM & LISA) as part of this work, so there should be no further dependence to closed source models.

---

### Official Review · Reviewer_UhN1 · 2023-08-05

**Soundness:** 3

**Excitement:**

3: Ambivalent: It has merits (e.g., it reports state-of-the-art results, the idea is nice), but there are key weaknesses (e.g., it describes incremental work), and it can significantly benefit from another round of revision. However, I won't object to accepting it if my co-reviewers champion it.

**Paper Topic And Main Contributions:**

In this article, authors propose a three stages approach to build a model that outputs a stylometric representation of a piece of text in a single forward pass. To do that, 1) they build a training dataset by generating stylistic description of texts with a decoder LM (GPT-3) 2) they use these to train a model to match stylistic attribute with the text (in a contrastive manner) 3) by downsampling the number of attributes with different heuristics, they train a regression model on the second model output probability. As attribute are Natural language description of style, the obtained dimensions are fully interpretable. The approach performs well on the STEL benchmark compared to prior work and correlates partially with human judgement

**Reasons To Accept:**

The examples of style attribute provided as examples seem indeed interesting for stylometric analysis of text corpora.
The method correlates well with human judgement for sentiment/emotion.
The authors propose to share de dataset and the model that output the style embedding. This would largely benefit the community
The experiments are convincing and appropriate/comprehensive.
The method is relatively new (even if quite incremental)


**Reasons To Reject:**

The output of SFAM does not correlate with human judgement on pure linguistic (which is what author are trying to achieve here, not building another sentiment analysis model).
Some details are missing to reproduce, e.g. for the experiment on Correlation to Human Judgement.


**Reproducibility:**

3: Could reproduce the results with some difficulty. The settings of parameters are underspecified or subjectively determined; the training/evaluation data are not widely available.

**Reviewer Confidence:**

3: Pretty sure, but there's a chance I missed something. Although I have a good feel for this area in general, I did not carefully check the paper's details, e.g., the math, experimental design, or novelty.

---

> ### Author Rebuttal · Authors · 2023-08-23
>
> We thank the reviewer for the comments.
>
> The review is correct that we find high correlation with tasks like sentiment, and we would like to highlight that we also find medium correlation with tasks like emotion detection, harmful language, and author profiling, all of which can contribute to writing style to some degree. In some of our examples, we demonstrate capturing some fairly interesting niche elements of writing style like "*The author is being polite.*" or "*The author is using a personal anecdote to illustrate their point.*", but there aren't many large datasets available for evaluating these kinds of fine-grained stylistic features so they are under-represented in our SFAM evaluation in Table 4. Through manual human annotation, however, we found that our overall error rate (18%) (Line 408,415,422) is fairly low. We believe this model does capture various linguistic properties and would be useful to the community for broad automated stylistic analysis, for which there are not many tools today.
>
> For reproducibility of the "Correlation to Human Judgements" experiments, we have submitted a code package with this submission, however, files too large to upload (datasets & model checkpoints), will be released with the final version of the paper on GitHub / HuggingFace Hub. We have, however, anonymously uploaded the code & data for this experiment here: [https://anonymous-drive.s3.amazonaws.com/sfam_eval.zip](https://anonymous-drive.s3.amazonaws.com/sfam_eval.zip) that should help alleviate any reproducibility concerns around these experiments.
>
> Overall, as this reviewer stated, we believe that our trained model is interesting to the community for wide-ranging stylometric analysis of text corpora and it would be one of the first models available to do so. This track is about "Stylistic Analysis" and we believe it's one of the first models that is capable of doing broader stylometric analysis over many dimensions of style.

---

### Official Review · Reviewer_9v8q · 2023-08-05

**Soundness:** 3

**Excitement:**

3: Ambivalent: It has merits (e.g., it reports state-of-the-art results, the idea is nice), but there are key weaknesses (e.g., it describes incremental work), and it can significantly benefit from another round of revision. However, I won't object to accepting it if my co-reviewers champion it.

**Paper Topic And Main Contributions:**

This paper aims to design an interpretable text representation (named LISA) that describes the style. To sum up, each dimension in their proposed representation vector corresponds to a style, where the value is obtained by running a style scorer (named SFAM). To select styles embedded by the representation and train the scorers, they prompt GPT-3 to describe and consolidate the styles in the MUD dataset that consists of Reddit posts from different users (StyleGenome). In their experiments, the authors find that their SFAM scorers are able to correlate well with human on some styles and the LISA embedding can perform comparably well with existing style embeddings on finding texts with similar styles (STEL evaluation framework). The contributions of this paper are:
- They design a framework for obtaining style annotations from LLMs, producing the StyleGenome dataset.
- They propose LISA, a text representation that measures intensity of different styles, thus interpretable.

**Questions For The Authors:**

- How similar are the styles of texts written by the same user? Though you show some sample reddit posts, are their styles consist across different posts?

**Reasons To Accept:**

- They introduce StyleGenome, a large-scale dataset containing 10000 reddit posts paired with all possible style attributes annotated by GPT-3.
- Their proposed LISA embedding is able to outperform existing style representations on some styles under the STEL framework.

**Reasons To Reject:**

- Despite correlating well with human on styles that are more semantics-relevant (e.g., sentiment, emotion), their SFAM scorers which serve as fundamental components of the LISA embedding do not correlate well with human on more linguistics-relevant styles (e.g., simplification, linguistic acceptability). This probably makes the LISA more content-focused, rather than style-focused.
- The authors try to avoid comparisons with content-aware representations, claiming those representations have direct access to the text and focus more on content (Line 349). However, as mentioned above, LISA is very likely to be content-aware. One thing the authors can do is checking what dimensions are contributing to the similarity measure. This could also evidence the interpretability. It is currently unclear whether the formality (or other style) dimension in LISA is really assisting finding texts that are similarly formal (or other style).

**Reproducibility:**

4: Could mostly reproduce the results, but there may be some variation because of sample variance or minor variations in their interpretation of the protocol or method.

**Reviewer Confidence:**

4: Quite sure. I tried to check the important points carefully. It's unlikely, though conceivable, that I missed something that should affect my ratings.

---

> ### Author Rebuttal · Authors · 2023-08-23
>
> We thank the reviewer for their comments. Below we respond to some of the "Reasons to Reject", however, we find the critiques in this review are misaligned with the actual content of the paper / factually incorrect and not in alignment with other reviewers. We provide the reasoning below and hope the review is re-considered in this light.
>
> **Reason 1:** The review states that our model correlates well with styles that are more semantics-relevant vs. linguistics-relevant, however, this dichotomy by the review doesn't make sense. Semantics is a *part of* linguistics; linguistics includes semantics [1]. If our model is capturing semantics, it is, by definition, capturing linguistic properties. Moreover, the review asserts "LISA is very likely to be content-aware" with no evidence and is directly contrary to our experimental results. We discuss and cite a number of works that find that content and style are inherently entangled and cannot be cleanly separated in many cases (Line 389, [2]). Therefore, *all* style representations will capture *some* content, even the prior style embeddings work by (Wegmann 2022) we compare to. How much style vs. content a model captures will inevitably fall on a continuous spectrum. We never claim otherwise and state it upfront (Line 389). In our discussion, we find that this percentage is reasonably low (3% of dimensions) (Line 389).
>
> **Reason 2:** We are not trying to avoid comparisons with content-aware representations and no other review has brought this as an issue. This paper is explicitly about training style representations that are content-independent. There is an entire discussion about this in the paper (Line 349). Content-aware representations like SBERT and LUAR are not what we are training in this paper; we are not trying to create a general-purpose sentence embedding model like SBERT. SBERT will produce a high cosine similarity for two sentences with different styles but similar content by the nature of its training, the exact opposite of the objective of this paper. Representations trained to specifically maximize capturing style and minimize capturing content, like LIWC and the style embeddings from (Wegmann 2022) are the appropriate baselines to compare against.
>
> **Unclear:** This review contains ungrammatical sentences like "This could also evidence the interpretability.", making it impossible for us to respond to these portions, could you clarify what you mean?
>
> [1] https://linguistics.ucla.edu/undergraduate/what-is-linguistics/
>
> [2] Style versus Content: A distinction without a (learnable) difference?: https://aclanthology.org/2020.coling-main.197/

---

### Meta-Review · Area_Chair_azMH · 2023-09-19

**Recommendation:** 3

**Metareview:**

The paper studies the task of learning interpretable style embeddings. To achieve this, the paper presents a synthetic data generation protocol from GPT-3 (using hand-crafted prompts) which is then used to train style representations. The style embeddings are shown to be competitive with recent approaches, while having a better interpretability. The use of LLM in the context of this goal is interesting and creative. I do share very similar concerns raised by cLTM, as the quality of the StyleGenome could have gone through a more rigorous evaluation, but the resource (to my opinion as the main contribution of this work) as-is has merits and potential for use by the community.

---

### Decision · Program_Chairs · 2023-10-07

**Decision:**

Accept-Findings

**Comment:**

The paper studies the task of learning interpretable style embeddings. To achieve this, the paper presents a synthetic data generation protocol from GPT-3 (using hand-crafted prompts) which is then used to train style representations. The style embeddings are shown to be competitive with recent approaches, while having a better interpretability. The use of LLM in the context of this goal is interesting and creative. I do share very similar concerns raised by cLTM, as the quality of the StyleGenome could have gone through a more rigorous evaluation, but the resource (to my opinion as the main contribution of this work) as-is has merits and potential for use by the community.